# Understanding high fructose corn syrup in popular snacks: Consumption, perceptions and labeling preferences

Oscar Sarasty[1]*, Modhurima Dey Amin[2]

**1** Ness School of Management and Economics, South Dakota State University, Brookings, South Dakota, United States of America, **2** Department of Agricultural and Applied Economics, Texas Tech University, Lubbock, Texas, United States of America

☯ These authors contributed equally to this work.
* Oscar.sarastymiranda@sdstate.edu

## Abstract

At a time when the Make America Healthy Again (MAHA) initiative brought renewed attention to High Fructose Corn Syrup (HFCS), this study explores the potential of supplemental labeling to inform consumers. Consumption of sugary items, including those containing HFCS, has been linked to non-communicable diseases like obesity in literature. We conduct a survey to examine respondents' perceptions and valuations of HFCS and their support for supplemental information labeling programs indicating the presence of HFCS, and the willingness to pay for HFCS labels in three food products. We find that 93% of consumers know what HFCS is, while over 58% report that all or most of the products they consume contain HFCS. Results from choice experiments suggest that consumers are willing to pay positive price premiums up to 16% of the average price for products with labels indicating the absence of HFCS. The findings suggest that supplemental HFCS labeling could serve as both a public health tool and a managerial policy lever—helping consumers make more informed choices while supporting broader efforts to reduce non-communicable diseases associated with excessive HFCS consumption. Additionally, the results provide food manufacturers with a strategic opportunity to align with evolving consumer preferences and potentially capture price premiums for HFCS-free products.

## Introduction

High fructose corn syrup (HFCS)—has re-emerged at the center of national discourse, fueled by political endorsement, corporate reformulation, and shifting consumer sentiment. In July 2025, President Donald Trump, through his "Make America Healthy Again" (MAHA) campaign, publicly declared HFCS "a formula for making you obese and diabetic," prompting speculation that Coca-Cola would eliminate the sweetener from its U.S. products [1–3]. The speculation follows a broader

**Data availability statement:** All relevant data are within the paper and its Supporting Information files.

**Funding:** The author(s) received no specific funding for this work.

**Competing interests:** The authors have declared that no competing interests exist.

trend: McDonald's recently replaced its Minute Maid lemonade with a "Premium Lemonade" made with real lemon juice and cane sugar, eliminating artificial additives and HFCS [4]. These highly publicized changes reflect intensifying consumer skepticism toward HFCS and growing preference for natural ingredients—primarily due to health concern [3,5,6].

Many food ingredients have been linked to rising obesity rates, and their manufacturing processes have also been associated with environmental impacts in the literature—particularly those with high concentrations of carbohydrates (e.g., [7,8]). Among these ingredients, sweeteners like sugar (from sugar cane) and HFCS (from corn) are frequently studied [8,9]. HFCS is a sweetener produced by converting corn starch into glucose [10]. It contains five percentage points more fructose than sugar [11], but is cheaper, has a liquid form, and offers a more intense flavor. As a result, it has been widely used in producing processed foods such as snacks and soft drinks, becoming a predominant caloric sweetener.

Consumption of both sugar and HFCS has increased over time (e.g., [12–15]). In the U.S., per capita HFCS consumption rose from 0 pounds in 1970–60 pounds in 2000, making up half of total annual sweetener intake [11]. Alarmingly, 18% of the U.S. population consumes twice the recommended amount of added sweeteners, much of it from ultra-processed foods high in HFCS [16]. The growth of HFCS consumption is particularly concerning due to its strong association with non-communicable diseases (NCDs), especially obesity [11,17]. This is a pressing public health issue: U.S. obesity rates increased from 30.5% to 42.4% between 1999 and 2018 and contribute to NCDs responsible for 71% of global deaths [18,19].

While all sweeteners have potential health implications, the primary difference between HFCS and sugar is their fructose content. HFCS is often considered a high-FODMAP sweetener because it contains a higher proportion of fructose relative to glucose. Excess fructose can be rapidly absorbed by the body and has been associated with inflammation, insulin resistance, increased fat production, and non-alcoholic fatty liver disease [20]. In addition, many individuals may have difficulty absorbing excess fructose, which can worsen symptoms such as bloating, abdominal pain, or post-meal discomfort, and may contribute to inflammatory conditions, including inflammatory bowel disease [21–23].

Given the public health relevance of HFCS and other sweeteners, this study investigates consumer perceptions and preferences for supplemental HFCS labeling on popular snack items: yogurt, granola bars, and honey wheat bread. These snacks were chosen because consumers often perceived them as a healthier alternative, despite some brands containing added sugars such as HFCS [24]. Previous literature (e.g., [25]) has examined consumers' preference for HFCS in soft drinks identifying negative attitudes; however, little is known about the presence of HFCS in products perceived as a healthier alternative potentially contributing to an unintentional overconsumption of added sugars [26]. Our specific objectives are (1) to identify consumer perceptions, consumption patterns, and purchasing behavior of products with HFCS; (2) to evaluate consumer preferences for different nutritional information

labels indicating the presence of HFCS; and (2) to determine the sociodemographic characteristics associated with consumer preferences for HFCS information labels.

At a time when the MAHA initiative has brought renewed attention to HFCS, the motivation of this research relies on the opportunity to understand the effect of a supplemental information label regarding the presence of HFCS. For example, sweeteners are already reported in the Food and Drug Administration's (FDA) mandatory labels on food products [10]. However, supplemental information on nutritional labels in processed food products that highlight the presence of sugar, fat, and salt may increase consumer awareness of the nutritional properties of the food product and increases the probability of selecting more healthy food options [27]. It is also important that supplemental information is supported by federal policy. For example, [28] have found that consumers better accept nutritional labels if the government approves the label claims, suggesting that a supplementary information food label indicating the presence of HFCS supported by the government would be better received by the general consumer.

The existing literature has not yet comprehensively addressed several pertinent issues regarding HFCS. It remains unclear how conscious consumers are of the presence of HFCS in their daily food items, whether they are informed about the associated health problems and environmental concerns, and the extent to which consumers actively look for HFCS presence on product labels and their preferences for HFCS-free products. In the U.S., the FDA is responsible for the labeling requirement of food products under the Federal Food, Drug, and Cosmetic Act. Food products must present two mandatory food labels: the general information label, which includes the units of measurement, manufacturer information, country of origin, ingredients, and allergens, and the nutritional label, which includes amounts of nutrients, calorie content, daily percentage value, and added ingredients [29].

Research on HFCS consumer perceptions and preferences is limited. However, existing literature provides valuable insights into how consumers perceive this sweetener relative to other sweeteners like sugar and how they make choices based on their understanding of its attributes and quality. Findings suggest that HFCS can be perceived to have a similar quality to sugar [30]. The literature has found that some products manufactured with HFCS can be seen as higher quality by consumers who lack knowledge of food quality information, and they are more likely to buy products that contain HFCS [31]. For example, [32] evaluated perceptions and acceptance of soft drinks made with different sweeteners. The study found that soft drinks with HFCS at 10% are perceived as the sweetest and least bitter sweetener compared with aspartame, sucralose, and acesulfame, making HFCS the preferred sweetener.

However, studies that assess consumer preference for HFCS are often secondary to other research objectives and are mostly limited to local samples. For instance, [25] examined consumer WTP for sweeteners in soft drinks. The study found a higher consumer acceptance of sweetened beverages made with sugar compared to those beverages made with HFCS. Also, their auction experiments revealed that consumers bid $0.39 lower for sweetened beverages that contain HFCS. Similarly, a study by [33] showed that consumers preferred baked goods without HFCS. In a more comprehensive study using phone surveys, [34] investigated consumer perception and preferences for HFCS in cereal, salad dressing, soft drinks, and yogurt. Results show that consumers perceive products with HFCS as less healthful and less expensive when HFCS is substituted for regular sugar. Results suggest that HFCS avoiders are willing to pay premiums of $0.55 for cereal, $0.43 for salad dressing, $0.14 for soft drinks, and $0.30 for yogurt.

Despite the literature presented above, a knowledge gap persists regarding consumer perceptions of HFCS and its influence on decision-making, especially in snack items. Acceptance and preference measurement for labels has been a significant focus in the literature, with many studies examining consumer preferences for various food labels that convey product attributes—such as USDA Organic, animal welfare, and local origin—or indicate the presence or absence of certain ingredients, like non-GMO labels (e.g., [35–37]). Moreover, studies such as [38] have explored consumer support for new labeling initiatives, finding, for instance, that consumers are willing to pay $134 annually for a mandatory GMO labeling program. Thus, delving into consumers' attitudes toward HFCS labeling and their health and environmental

perceptions of HFCS could significantly inform policy decisions and provide insights into consumer support for potential new labeling programs.

The study contributes to understanding consumers' perceptions and demand for HFCS, which can eventually lead to programs that improve population nutrition and reduce obesity. Moreover, the results from the study seek to unveil how ingredients transparency can correct product misinterpretation and potentially guide to healthier food choices in products that can be overlooked. The findings also aim to initiate managerial policy discussion regarding the premium of HFCS reporting and the potential gain from supplemental HFCS information on food labels. The insights gained from understanding these influences can guide policymakers in outlining effective HFCS labeling policies.

## Methods

We developed a survey instrument that was divided into three sections to collect information on 1) informed consent and sociodemographic characteristics, 2) respondents' perception, and consumption of HFCS products, and 3) a set of choice experiments to assess respondents' preferences for different HFCS labels. Participants that agreed to an informed consent were able to complete the survey.

Ethical clearance was obtained from the Institutional Review Board of Texas Tech University (Approval no. IRB2021−972). The study protocol and survey instrument were reviewed and approved prior to data collection. Participants provided informed consent electronically before beginning the survey, and they were informed that participation was voluntary, responses would remain anonymous, and they could withdraw at any time. Data were collected from January 24, 2022, until June 17, 2022, and analyzed shortly after data collection was completed. All identifiers were removed before the formal analysis.

### Choice experiments

Choice experiments (CE) and contingent valuation are two survey experiment approaches commonly used to measure consumer preferences and willingness to pay (WTP) for hypothetical product attributes. While both approaches present unique strengths and limitations, we opted for CE given that we can get respondents to make decisions in a more realistic decision scenario that can simulate the real life decision process for hypothetical products compared to contingent valuation. It also allows us to quantify the respondent trade-offs for different attributes across multiple scenarios [39]. CE scenarios replicate situations in which respondents observe products differing in attributes and prices to measure WTP based on stated preferences (See Fig 1).

The products used for the CE included vanilla yogurt, granola bars (12 units), and honey wheat bread. These are non-liquid, everyday snack items that frequently contain HFCS and are commonly referenced in the labeling literature (e.g., [34]). In the choice scenario, respondents were asked to choose their preferred product profile between two items with identical flavor and presentation characteristics but varying prices and in the presence or wording of the HFCS label attribute. Respondents were likely to face choice scenarios where both products have no presence of HFCS. In that case, respondents selected their favorite product based on the wording of HFCS labels and the price attribute. Respondents who did not favor either option could select a "none" option.

The attributes used in the CE were HFCS labels and price. For the HFCS attribute, the base level was the absence of a label. Additional levels included a label indicating the presence of HFCS (Contains High Fructose Corn Syrup), and three different phrasing indicating its absence: "No High Fructose Corn Syrup", "Free of High Fructose Corn Syrup", and "Does not contain High Fructose Corn Syrup". Using three different wordings—"No High Fructose Corn Syrup," "Free of High Fructose Corn Syrup," and "Does not contain High Fructose Corn Syrup"—adds value by allowing us to assess whether subtle variations in label phrasing influence consumer perception and willingness to pay. When presented alone, consumers may interpret these claims differently in terms of credibility, clarity, or implied health benefits. By including multiple absence claims, we can isolate whether specific wording elicits stronger preference or skepticism, helping policymakers and marketers better understand how phrasing shapes consumer responses and perceived product value.

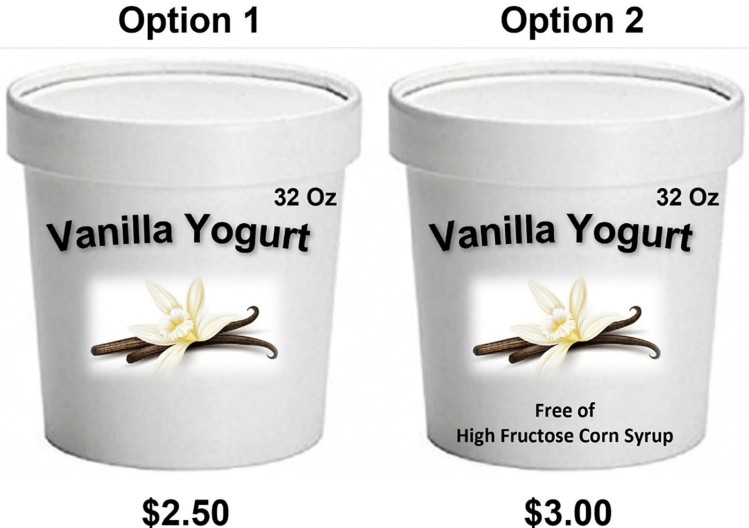

**Fig 1. Example of a choice experiment.** Note: Fig 1 presents a sample choice scenario from the experiment, where respondents were shown product images with varying HFCS label claims (e.g., no label vs. "Free of High Fructose Corn Syrup") and asked to select their preferred option. These visual stimuli were used to simulate realistic product comparisons and assess willingness to pay for labeling attributes. See Methods section for experimental design details.

The price attribute consisted of four different prices for each product. These prices were randomly defined based on a sample range of products with and without HFCS collected from online retailers. Table 1 describes the products and attributes displayed in the CE. The first column indicates the product, the second column presents the price levels for each product, the third details the HFCS labels used in the choice experiments, and the fourth provides the average market price, including minimum and maximum prices. To prevent choice scenarios where respondents encounter the same attribute (two products claiming the absence of HFCS), the wording that indicated the absence of HFCS in the product was randomly assigned.

Demand theory suggests that consumers make purchase decisions by selecting product attributes that maximize their utility within a budget constraint. Our focus is on HFCS labels as the attribute of interest. The econometric model builds on Lancaster's characteristics-based demand theory and McFadden's [38] random utility model (RUM) [40,41]. A respondent's indirect utility function is influenced by product attributes, including the price [42]. For respondent j facing choice set s, the utility of alternative $l$ is:

$$U_{jsl} = V_{jsl} + e_{jsl} \tag{1}$$

where $j = 1,…,J$ indicate respondents, $l = 1,…,L$ show alternatives, and $s = 1,…,S$ represent choice sets. The variable $V_{jsl}$ represents observable utility from price and non-price attributes, while $e_{jsl}$ captures unobserved factors. We specify $V_{jsl}$ as linear in price $p_{jsl}$ and the non-price attributes $X_{jsl}$:

$$U_{jsl} = -\alpha_j p_{jsl} + \beta_j' X_{jsl} + e_{jsl} \tag{2}$$

where, the parameter $\alpha_j$ captures the marginal disutility of price and $\beta_j'$ represents the marginal utilities received from attributes.

To estimate the willingness-to-pay (WTP) directly, we re-parameterize utility following [42]. Define $\theta_j = \begin{bmatrix} \gamma_j & Z_j' \end{bmatrix}'$ where $\gamma_j = \alpha_j$ and $Z_j$ is a vector of WTP values for attributes. Systematic utility becomes:

**Table 1. Products and attributes of choice experiments.**

| Product | Price attribute | HFCS labels attribute | Market price |
|---|---|---|---|
| Vanilla yogurt (32 Oz) | $2.10<br>$2.50<br>$2.75<br>$3.00 | No label<br>Contains HFCS<br>Free of HFCS<br>Does not contain HFCS<br>No HFCS | Average: $2.60<br>Minimum: $1.84<br>Maximum: $4.99 |
| Granola bars (12 Units) | $2.00<br>$2.35<br>$2.65<br>$2.90 | | Average: $2.50<br>Minimum: $1.14<br>Maximum: $3.58 |
| Honey wheat bread (20 Oz) | $1.75<br>$2.10<br>$2.35<br>$2.55 | | Average: $2.22<br>Minimum: $0.98<br>Maximum: $4.48 |

Note: The table summarizes the products, price levels, and HFCS-related labeling attributes used in the choice experiments. Price levels were selected to reflect realistic variation around the market average, based on actual retail prices collected from major U.S. grocery retailers in 2024. The HFCS labeling attribute includes both absence claims ("Free of HFCS," "No HFCS," "Does not contain HFCS") and a presence label ("Contains HFCS") to examine consumer responses to different label formats. The "No label" option serves as the baseline. All participants saw randomized combinations of these attributes across multiple choice scenarios.

$$V_{jsl}(\theta_j) = -\gamma_j p_{jsl} + (\gamma_j Z_j)' X_{jsl} \tag{3}$$

Assuming $\epsilon_{jsl}$ is i.i.d. extreme value, the probability of respondent j choosing alternative l in set s, conditional on $\theta_j$, is [43]:

$$P_{jsl}(\theta_j) = \frac{\exp[V_{jsl}(\theta_j)]}{\sum_{n=1}^{L} \exp[V_{jsn}(\theta_j)]} \tag{4}$$

The probability of observing respondent $j$'s sequence of choices across all $S$ sets is:

$$R_j(\theta_j) = \Pi_{s=1}^{S} P_{jsl(s)}(\theta_j) \tag{5}$$

where $l(s)$ denotes the alternative chosen in set s [41]. Since $\theta_j$ is unobserved and varies across respondents with density $g(\theta_j|\Gamma)$, the unconditional choice probability (mixed logit) is:

$$P_j(\Gamma) = \int R_j(\theta_j) g_j(\theta_j|\Gamma) d\theta_j \tag{6}$$

The log-likelihood for all respondents is:

$$\log L(\Gamma) = \sum_{j=1}^{J} \ln[P_j(\Gamma)] \tag{7}$$

The parameters in Γ (such as the mean and standard deviation of the price coefficient and each WTP distribution) are estimated via simulated maximum likelihood in Stata [44,45]. We assume $\gamma_j$ and each element of $Z_j$ are normally distributed across respondents. Once the model is estimated, we can directly recover the mean WTP for each attribute from the estimates of Γ. For example, if $Z_{jk}$ represents the WTP for attribute $k$, the mean WTP for that attribute is $E(Z_{jk})$, obtained from the estimated distribution.

## Factors that affect WTP

After estimating the mixed logit model, we conducted a two-step analysis to explore how individual respondent characteristics relate to their WTP for the HFCS labeling attributes [46]. In the first step, we used the mixed logit results to compute individual-specific WTP estimates for each HFCS label attribute. In the second step, we regressed those WTP estimates on respondent-level factors (e.g., sociodemographic variables) to identify significant influences on WTP.

In the first step, we derive each respondent's individual WTP for the HFCS-related attributes by applying Bayes' rule to the mixed logit model results. Essentially, we obtain the posterior distribution of each respondent's preference parameters ($\theta_j$) given their observed choices. The density of $\theta_j$ conditional on respondent j's choices and parameters Γ is [43,47]:

$$f(\theta_j|\Gamma) = \frac{R_j(\theta_j) \cdot g(\theta_j \mid \Gamma)}{P_j(\Gamma)}$$

(8)

Here, $R_j(\theta_j)$ is the likelihood of j's observed choice sequence given a specific parameter vector, as defined in equation (5) above, $g(\theta_j \mid \Gamma)$ is the prior density of $\theta_j$ in the population, and $P_j(\Gamma)$ is the mixed logit choice probability. From this posterior density, we can obtain the expected value (posterior mean) of $\theta_j$ for each respondent.

$$E[\theta_j \mid \Gamma] = \int \theta_i f(\theta_i \mid Y_j, \Gamma) \, d\theta_j$$

(9)

Because the integral in (9) does not have a closed form, we use simulation to approximate this expectation. Specifically, we draw $q = 1, \ldots, Q$ random draws of $\theta$ from $g(\theta \mid \Gamma)$, then weight each draw by its likelihood of producing respondent $j$'s observed choices. The simulated posterior mean of $\theta_j$ can be written as:

$$\hat{E}[\theta_j \mid \Gamma] = \frac{\sum_q \theta_j^{(q)} R_j\left(\theta_j^{(q)}\right)}{\sum_q R_j\left(\theta_j^{(q)}\right)}$$

(10)

where $\theta_j^{(q)}$ is the $q^{th}$ random draw of $\theta$ for respondent $j$ from the population distribution, and $R_j(.)$ is the likelihood of j's observed choices under that draw. In essence, equation (10) is a weighted average of the sampled θ's, where draws that better predict the respondent's choices receive higher weight [48]. In our estimation, we used the maximum likelihood estimates $\hat{\Gamma}$ in place of the true Γ when drawing from the population distribution (a standard practice in simulated posterior calculations; see [47]). We experimented with different numbers of draws $Q$ to ensure stability of the individual WTP estimates and found that 1,000 draws were sufficient for convergence. Thus, for each respondent we obtained estimates of their WTP for the HFCS presence label and for each HFCS absence-claim wording. These individual WTP estimates (one for each attribute level of interest per respondent) became the dependent variables for the second-step analysis.

In the second step, we analyze how respondents' characteristics influence their WTP for the HFCS labeling attributes [49]. Because each respondent contributes multiple WTP observations (one for each label type), we employ a panel data regression model with random effects to account for the repeated observations per individua. In particular, we estimated a random-effects linear regression of the form:

$$WTP_{jc} = h'_j b + u_j + \epsilon_{jc}$$

(11)

where $WTP_{jc}$ is the estimated WTP of respondent $j$ for attribute (label) c, $h_j$ is a vector of respondent's observed characteristics, $b$ is the corresponding coefficient vector, $u_j$ is a respondent-specific random effect, and $\epsilon_{jc}$ is the idiosyncratic error term. The coefficients in $b$ provide the average marginal effects of respondent characteristics on WTP for the

HFCS-related labels. This model pools the WTP for all HFCS label attributes together while allowing for within-respondent correlation [49]. We estimated this WTP regression in Stata.

Finally, we took precautions to mitigate hypothetical bias in our CE, since respondents might overstate their WTP in a hypothetical setting. Following best practices in stated preference studies, we implemented a cheap talk script prior to the choice questions. After the information treatment and just before the first choice scenario, respondents read a brief statement urging them to imagine they were actually shopping and spending their own money, and to answer as realistically as possible. This cheap talk script reminds participants about the tendency to overstate valuations in surveys and explicitly asks them to avoid doing so. By making respondents conscious of hypothetical bias, cheap talk can significantly reduce overestimation of WTP in stated preference elicitation [50]. In our survey, the inclusion of the cheap talk prompt was intended to calibrate respondents' mindsets, helping to produce WTP estimates that better reflect real-world behavior.

## Results

### Data and descriptive statistics

We collected the data through an online survey administered to U.S. households via Amazon Mechanical Turk, a widely used crowdsourcing platform for participant recruitment. After applying a screening criterion—where respondents needed to be at least 18 years old and in charge of their food purchases—and removing incomplete surveys, 1,006 observations remained available for statistical analysis. The primary sociodemographic features of the sample closely align with those reported for the broader U.S. population in the most recent U.S. Census. The average age of respondents was around 37 years (compared to the U.S. average of 38.5 years), 81% have college education or more, about half of them were male (mirroring the 49% US average), and 64% were married. A 74% of respondents identified as White (compared to 76.3% in the U.S. population), and 29% report being Hispanic or Latino background. The average household characteristics present a size of 3.2 (almost identical to the U.S. average), and the average income was about $60 thousand a year. Table 2 presents a detailed breakdown of the sociodemographic characteristics of the sample.

Table 3 presents the respondents' perceptions and consumption of HFCS. Aiming to understand if consumers know what HFCS is, we included a self-knowledge question asking "Do you know what HFCS" where nearly 93% of survey participants claimed they know what HFCS is. To identify what is the perception of what products contain HFCS, we asked them to select the products that the think have HFCS from a list of products that usually are made with HFCS. Among the products that more than half of respondents consider have HFCS, we identified candies (71%), soft drinks (69%), juice drinks (63%), packed sweets like cupcakes and pastries (58%), and snacks (51%). All these identified products contain HFCS and are available on the market [8,12,51]. Furthermore, when asked about their HFCS consumption, 18% of the respondents noted that every product they consume contains HFCS, and about 78% report that a significant portion of the products they frequently consume have HFCS. Notably, approximately 88% of respondents expressed that products with HFCS should carry a mandatory front label.

### Mixed logit model results

Results from the mixed logit estimations of utility in the preference space are presented in Table S1 Table. Separate models were estimated for each product—vanilla yogurt, granola bars, and honey wheat bread—featured in the choice experiment scenarios. The findings indicate that respondents derive additional utility from HFCS-free labeling compared to the baseline of no label. In particular, labels such as "Free of High Fructose Corn Syrup" and "Does Not Contain High Fructose Corn Syrup" are associated with positive and statistically significant coefficients, suggesting a clear preference for these phrasings. Conversely, the "Contains High Fructose Corn Syrup" label consistently enters with a negative and significant coefficient across all products, indicating disutility from explicit HFCS disclosure. The "No High Fructose Corn

**Table 2. Sociodemographic characteristics of the sample.**

| Variable | Mean (Std. Dev) |
|---|---|
| Age | 36.963 (11.955) |
| College graduated or more | 0.810 (0.392) |
| Sex of the respondent | |
| Male | 0.518 (0.500) |
| Female | 0.482 (0.500) |
| Married | 0.640 (0.480) |
| Race of the respondent | |
| White | 0.739 (0.440) |
| Other | 0.261 (0.241) |
| Hispanic or Latino Background | |
| Hispanic or Latino | 0.287 (0.423) |
| Non-Hispanic or Latino | 0.713 (0.453) |
| Household characteristics | |
| Household size | 3.201 (1.234) |
| Household with minors | 0.634 (0.482) |
| Annual household income (thousand $) | 60.509 (41.803) |

Note: The table presents summary statistics for the sociodemographic characteristics of the survey sample (n = 1,006). Means are reported for continuous variables, while proportions (and corresponding standard deviations) are shown for binary indicators. The sample is broadly representative of U.S. adults across key demographics, including age, education, race/ethnicity, household composition, and income. See the Data section for further details on sampling procedures, variable construction, and survey implementation.

Syrup" label, however, was not significant in two of the three models, pointing to weaker or more ambiguous consumer interpretation. Price coefficients are negative and statistically significant in all models, consistent with economic theory that higher prices reduce consumer utility. The alternative-specific constants (ASCs) are also large, negative, and significant, suggesting that respondents generally preferred selecting one of the product options over the opt-out alternative. Finally, many of the standard deviation estimates for the random coefficients are statistically significant, confirming substantial heterogeneity in consumer preferences for HFCS labels and price across the sample.

Table 4 presents the estimated marginal WTP values, calculated using the coefficients from S1 Table. These estimates represent the average price premiums or discounts that consumers assign to products with specific HFCS-related labels, relative to an unlabeled baseline. Consistent with the preference-space results, respondents exhibit a positive WTP for labels indicating the absence of HFCS and a negative WTP for labels indicating its presence. These findings highlight that ingredient information on food labels significantly influences consumer preferences. Negative WTP for HFCS-containing labels suggests that consumers are not only less willing to purchase such products but may actively avoid them. This pattern aligns with prior literature on nutritional labeling. For example, [27] reviewed 120 peer-reviewed studies and found that supplemental information labels can shift consumer choices toward healthier products and reduce WTP for products perceived as less healthful.

For vanilla yogurt, the pattern of premiums closely follows the aggregate results. Consumers are willing to pay about $1.24 more for a yogurt labeled "Does Not Contain High Fructose Corn Syrup" and $1.22 more for "Free of High Fructose Corn Syrup," both significant at the 1 percent level. By contrast, the phrasing "No High Fructose Corn Syrup" does not command a significant premium, while the presence label "Contains High Fructose Corn Syrup" reduces WTP by roughly $0.79. Using the sample's average shelf price of $2.60, a "Free of HFCS" variant would be valued at approximately $3.82—still below the highest observed market price of $4.99. These results are similar to [34], who found that HFCS

**Table 3. High Fructose Corn Syrup (HFCS) perceptions and consumption.**

| Variable | Percentage |
|---|---|
| Do you know what HCFS is? | |
| Yes | 93.33 |
| No | 6.67 |
| What are the products you think have HFCS? [a] | |
| Candy | 71.07 |
| Soft drinks | 68.69 |
| Juice drinks | 63.12 |
| Packaged Sweets (cupcakes, pastries, cookies, etc.) | 57.95 |
| Snacks | 50.89 |
| Ice cream and ice pops | 46.02 |
| Sauces and condiments | 46.22 |
| Breakfast food items | 38.57 |
| Bread and crackers | 31.71 |
| Peanut butter | 26.24 |
| Cold cuts | 17.69 |
| Other | 1.09 |
| What is the consumption of HFCS? | |
| All the products I consume have HFCS | 17.51 |
| Most of the products I consume have HFCS | 30.85 |
| I frequently consume products with HFCS | 27.06 |
| I try to avoid products with HFCS | 18.31 |
| I never consume products with HFCS | 2.59 |
| I don't know/refuse to answer | 3.68 |
| Do you think products with HFCS should have a mandatory front label? | |
| Yes | 87.67 |
| No | 12.33 |
| [a]Respondents were able to select multiple answers | |

Note: The table summarizes respondents' awareness, perceived sources, and self-reported consumption of HFCS, along with views on mandatory front labeling (n = 1,006). Multiple responses were allowed for product categories. See the Data section for details.

disclosures lower yogurt valuations, and they stand in contrast to [52], whose 6-oz yogurt study detected only a negligible $0.07 premium for a "Free of HFCS" claim; when scaled to a 32-oz package, our estimate ($1.22, or about $0.22 per 6 oz) is materially larger, indicating stronger consumer responsiveness in the present sample.

A similar pattern emerges for granola bars, with consumers expressing significant premiums for labels that signal the absence of HFCS. The average WTP is $0.93 for "Free of High Fructose Corn Syrup" and $0.62 for "Does Not Contain High Fructose Corn Syrup," both statistically significant. As in the yogurt case, the label "No High Fructose Corn Syrup" does not yield a statistically significant premium. Meanwhile, the presence label "Contains High Fructose Corn Syrup" reduces WTP by $0.42. These findings are consistent with [31], who observed that consumers tend to avoid granola products explicitly labeled as containing HFCS.

For honey wheat bread, the patterns again hold. The "Free of High Fructose Corn Syrup" label commands the highest premium at $1.01, followed by $0.70 for "Does Not Contain High Fructose Corn Syrup," and $0.28 for "No High Fructose

**Table 4. Estimated Marginal Willingness-to-Pay Values.**

| Attribute | WTP Calculation | Mean WTP | | 95% Confidence Interval for the Mean |
|---|---|---|---|---|
| Vanilla yogurt | | | | |
| Does not contain HFCS | $-(\beta_{Does\ not\ contain\ HFCS}/(\beta_{price}))$ | 1.236 | *** | 0.618~1.854 |
| Free of HFCS | $-(\beta_{Free\ of\ HFCS}/(\beta_{price}))$ | 1.224 | *** | 0.772~1.677 |
| No HFCS | $-(\beta_{No\ GFCS}/(\beta_{price}))$ | 0.010 | | −0.516~0.535 |
| Contain HFCS | $-(\beta_{Contain\ HFCS}/(\beta_{price}))$ | −0.788 | ** | −1.426~−0.151 |
| Granola bars | | | | |
| Does not contain HFCS | $-(\beta_{Does\ not\ contain\ HFCS}/(\beta_{price}))$ | 0.620 | *** | 0.241~0.999 |
| Free of HFCS | $-(\beta_{Free\ of\ HFCS}/(\beta_{price}))$ | 0.925 | *** | 0.650~1.199 |
| No HFCS | $-(\beta_{No\ GFCS}/(\beta_{price}))$ | 0.722 | | −1.323~2.777 |
| Contain HFCS | $-(\beta_{Contain\ HFCS}/(\beta_{price}))$ | −0.423 | ** | −0.891~−0.014 |
| Honey Wheat Bread | | | | |
| Does not contain HFCS | $-(\beta_{Does\ not\ contain\ HFCS}/(\beta_{price}))$ | 0.699 | *** | 0.299~1.099 |
| Free of HFCS | $-(\beta_{Free\ of\ HFCS}/(\beta_{price}))$ | 1.011 | *** | 0.645~1.378 |
| No HFCS | $-(\beta_{No\ GFCS}/(\beta_{price}))$ | 0.278 | | −0.508~1.064 |
| Contain HFCS | $-(\beta_{Contain\ HFCS}/(\beta_{price}))$ | −1.071 | ** | −1.943~−0.199 |

Note: ***indicates significance at 1%, ** indicates significance at 5%, and * indicates significance at 10%. HFCS is the acronym of "High Fructose Corn Syrup"

Corn Syrup," though the latter is not statistically significant. Bread bearing the "Contains High Fructose Corn Syrup" label incurs a WTP discount of $1.07, the largest negative valuation across all three product categories. These consistent WTP patterns reinforce the robustness of the consumer aversion to HFCS and the signaling value of its absence across product types.

The uniformly negative WTP estimates for the "Contains High Fructose Corn Syrup" label reinforce earlier evidence that consumers regard HFCS—or added sugars more generally—as undesirable. For example, [34] showed that perceived healthfulness scores for granola and yogurt dropped from 6.03 and 5.91 (on a 1–9 scale) to 4.02 and 4.19 when the products were labeled as containing HFCS. Likewise, [53] reported that explicit "added-sugar" disclosures prompt consumers to avoid those items and favor alternatives carrying "free" or "does not contain" claims—paralleling our positive WTP for HFCS-free phrasings. Together, these findings point to a shared public sentiment: ingredient transparency about HFCS (or sugar) leads many shoppers to discount the labeled products and seek lower-sugar options. At the same time, prior work also cautions that labeling effects can vary by product context; for indulgent categories such as ice cream, [53] found that some consumers preferred the information be absent, mirroring our nonsignificant result for the "No HFCS" phrase in certain processed items. Finally, [28] documented consumers' preference for concise, straightforward labels—an insight echoed here, where clearer wording ("Free of HFCS" or "Does Not Contain HFCS") garnered stronger positive valuations than more ambiguous phrasing.

S2 Table reports the results of random-effects regressions assessing whether individual-level sociodemographic characteristics explain variation in WTP for HFCS labels across the three product categories. Most coefficients are statistically insignificant, including those for age, gender, education, income, and household composition. This suggests that preferences for HFCS-related labels—and the premiums consumers are willing to pay—are not strongly driven by observable demographics, but rather reflect heterogeneous preferences across the population. A few exceptions emerged: Hispanic or Latino respondents displayed a statistically significant higher WTP for vanilla yogurt, and larger household size was positively associated with WTP for granola bars. However, these isolated effects do not reflect consistent demographic trends across products. The ρ (rho) values, which range from 0.149 to 0.265, indicate modest intra-individual correlation

in WTP across product categories, further supporting the presence of individual-level taste heterogeneity in consumer responses to labeling.

## Discussion

The findings offer meaningful implications for both industry and policy. From a policy standpoint, the evidence indicates that consumers are not only aware of potential health risks associated with HFCS but are also willing to pay a premium to avoid it. Nearly half of the respondents consistently avoided products containing HFCS, and statistically significant preferences were observed for items labeled as HFCS-free. These results suggest that front-of-package labels explicitly stating the presence of HFCS could enhance transparency and better align with consumer demand for ingredient disclosure. Such labeling may serve as an effective behavioral nudge, supporting public health efforts to reduce excessive caloric intake and the incidence of diet-related NCDs. These findings also align with the recent public health efforts from the MAHA initiative, which seeks to improve dietary choices through transparent food packaging, where supplemental labels can be a relevant policy tool that may contribute to reducing the consumption of HFCS.

The results also underscore that the effectiveness of supplemental labels depends not only on the disclosure of key nutritional information—such as the presence or absence of HFCS—but also on how that information is framed. The observed variation in consumer WTP across different phrasings ("Free of HFCS," "Does Not Contain HFCS," and "No HFCS") suggests that wording matters: while the first two consistently generated statistically significant positive WTP, the latter did not. This finding aligns with [54], who demonstrated that subtle shifts in label language can significantly affect consumer valuation, even when conveying identical product attributes. Similarly, [55] reported that sugar-related claims can elicit diverse consumer interpretations. These results highlight an important opportunity for regulators: setting clearer standards for label language could reduce ambiguity, prevent misleading impressions, and enhance consumer understanding. From a regulatory perspective, these insights suggest that wording is needed not only on what information should be disclosed, but also on how it should be communicated to maximize consumers' understanding of the food product. For food policy initiatives aimed at front-of-package labeling, adopting more effective and intuitive phrasing—such as "Free of HFCS" or "Does Not Contain HFCS"—could strengthen the impact of such disclosures on consumer behavior.

Furthermore, these findings carry managerial implications for the food industry. They suggest that manufacturers could benefit from highlighting the absence of HFCS in their products, as U.S. consumers demonstrate a measurable willingness to pay a premium for such claims. This premium could help offset potential costs associated with reformulating products by replacing HFCS with natural or less controversial sweeteners. Supporting this perspective, [56] found that HFCS operates in a price-sensitive market—estimating that a 1% increase in HFCS price leads to a 0.153% decrease in deliveries—underscoring how even modest shifts in price or demand can influence supply dynamics. In addition to reformulation, manufacturers can leverage these findings to enhance marketing strategies. Emphasizing the absence of HFCS could appeal to health-conscious consumers, helping brands differentiate themselves in a crowded marketplace and tap into a growing segment that prioritizes ingredient transparency and perceived healthfulness. Coca-Cola's recent decision to introduce a sugar cane version on its beverages by fall 2025 suggests that political influence and shifting consumer preferences can drive industry reformulation decisions. As the debates around added sweeteners may intensify under MAHA initiatives, industry reaction may include reformulation, where the use of supplemental labeling can serve as a strategic marketing tool as it also aligns with public health objectives.

## Conclusions

We examined consumer perceptions, consumption patterns, valuations, and their determinants related to HFCS labeling. The data were collected through an online survey administered to U.S. consumers. Results indicate that 93% of respondents are familiar with HFCS and can identify products that typically contain it, such as candy and soft drinks. HFCS appears to be widely consumed—48% of respondents reported that all or most of the products they consume contain

HFCS—although 18% actively try to avoid such products. The choice experiment revealed a consistent WTP premium for labels indicating the absence of HFCS. On average, consumers were willing to pay approximately 14% more for food products supporting mandatory HFCS labeling programs. Additionally, respondents displayed uniform aversion to products labeled as containing HFCS across all product categories included in the study. The lack of significant demographic effects on WTP suggests that preferences for HFCS labeling stem more from individual-level taste than from observable characteristics.

The study faced several limitations. First, we intentionally focused on three food products—yogurt, granola bars, and honey wheat bread—that are generally perceived as healthier choices and not typically associated with HFCS use. Given the wide range of products that contain HFCS, our findings may not be generalizable to products more commonly known to include it. Second, the survey design included multiple products and several rounds of choice sets, which may have led to cognitive fatigue among respondents, as suggested in prior research [46]. Finally, the study concentrated on consumer-side evaluations of HFCS without incorporating potential substitution effects with alternative sweeteners or producers' preferences. Future work could address these gaps by examining manufacturer incentives around HFCS use, leveraging real purchase data, or exploring how HFCS labels interact with other claims such as USDA Organic or "low sugar."

In conclusion, our findings suggest that HFCS labeling can serve as a feasible policy lever to support food choices, increase transparency, and encourage reformulation within the food industry. Importantly, the results also underscore that consumer response is shaped not only by the presence of nutritional information but also by how that information is framed. This highlights the strategic role of language in guiding dietary behavior and reducing the incidence of non-communicable diseases. As regulatory bodies, such as the FDA, continue refining the nutritional labeling framework—such as through revisions to the Nutrition Facts Panel—they may consider incorporating HFCS as a labeled ingredient of concern. Moreover, broader policy discussions surrounding sugar taxation, HFCS regulation, and incentives for natural sweeteners can benefit from empirical evidence of consumer aversion to HFCS, as reflected in the WTP estimates generated in this study.

## Supporting information

**S1 Table. Mixed Logit Model Estimation.** This supporting information table reports the estimated coefficients, standard errors, and significance levels from the mixed logit specification used to evaluate consumer preferences.
(DOCX)

**S2 Table. Random Effects Regression for sociodemographic characteristics.** This supporting information table presents results from the random-effects model linking sociodemographic characteristics with willingness to pay values.
(DOCX)

**S1 Data. Consumer choice experiment dataset.** This file contains the anonymized dataset used for the mixed logit and random-effects analyses, including choice outcomes, product attributes, prices, and sociodemographic variables. The data are provided in a compressed (RAR) format.
(RAR)

## Author contributions

**Conceptualization:** Oscar Sarasty, Modhurima Dey Amin.

**Data curation:** Oscar Sarasty.

**Formal analysis:** Oscar Sarasty.

**Funding acquisition:** Oscar Sarasty.

**Investigation:** Oscar Sarasty.

**Methodology:** Oscar Sarasty, Modhurima Dey Amin.

**Project administration:** Oscar Sarasty.

**Resources:** Oscar Sarasty.

**Software:** Oscar Sarasty.

**Supervision:** Oscar Sarasty, Modhurima Dey Amin.

**Validation:** Oscar Sarasty, Modhurima Dey Amin.

**Visualization:** Oscar Sarasty.

**Writing – original draft:** Oscar Sarasty, Modhurima Dey Amin.

**Writing – review & editing:** Oscar Sarasty.

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
