## [Decision Letter · Decision Letter 0]

19 Nov 2025

Dear Dr. Sarasty,

Thank you for submitting your manuscript to PLOS ONE. After careful consideration, we feel that it has merit but does not fully meet PLOS ONE’s publication criteria as it currently stands. Therefore, we invite you to submit a revised version of the manuscript that addresses the points raised during the review process.

We look forward to receiving your revised manuscript.

Kind regards,

Edwin Hlangwani, PhD

Academic Editor

PLOS ONE

**Journal Requirements:**

2. In the online submission form you indicate that your data is not available for proprietary reasons and have provided a contact point for accessing this data. Please note that your current contact point is a co-author on this manuscript. According to our Data Policy, the contact point must not be an author on the manuscript and must be an institutional contact, ideally not an individual. Please revise your data statement to a non-author institutional point of contact, such as a data access or ethics committee, and send this to us via return email. Please also include contact information for the third party organization, and please include the full citation of where the data can be found.

Reviewers' comments:

Reviewer's Responses to Questions

**Comments to the Author**

1. Is the manuscript technically sound, and do the data support the conclusions?

Reviewer #1: Yes

Reviewer #2: Yes

2. Has the statistical analysis been performed appropriately and rigorously?

Reviewer #1: Yes

Reviewer #2: Yes

3. Have the authors made all data underlying the findings in their manuscript fully available?

Reviewer #1: Yes

Reviewer #2: Yes

4. Is the manuscript presented in an intelligible fashion and written in standard English?

Reviewer #1: Yes

Reviewer #2: Yes

Reviewer #1: The authors have addressed a very timely topic in this manuscript, at a time when nutritional challenges such as high fructose corn syrup (HFCS) are increasingly recognized as detrimental for health. They have supported the consumer perception, consumption pattern, and valuations with crisp data.

The following comment may be incorporated to improve the manuscript further: When discussing about concerns due to consumption of HFCS, the authors may include a statement on HFCS in relation to FODMAPs and their impact on GI motility disorders such as IBS, post prandial distress, as well as inflammatory conditions such as IBD. These conditions afflict a lot of people around the world.

Reviewer #2: This study explores the links between high-fructose corn syrup (HFCS) consumption, public perception, and labeling preferences for popular snacks. The findings offer valuable insights for food manufacturers, policymakers, and health educators working to address the divide between the science of HFCS and consumer views. The methods are well-designed, and the results are clearly presented, contributing meaningful understanding to this often-debated ingredient.

**Do you want your identity to be public for this peer review?** For information about this choice, including consent withdrawal, please see our Privacy Policy

Reviewer #1: No

Reviewer #2: No

---

## [Author Response · Author response to Decision Letter 1]

19 Dec 2025

Dr. Edwin Hlangwani

Academic Editor

PLOS ONE

Dear Editor,

We would like to thank the Editor and reviewers for the time spent reviewing this manuscript and the important suggestions. After revising all the comments provided by the reviewers, the following changes in the manuscript were addressed:

Editor Comments

Response: We thank the editor for this comment. We made changes to the manuscript to follow the PLOS ONE style requirements, including those for file naming.

2. In the online submission form you indicate that your data is not available for proprietary reasons and have provided a contact point for accessing this data. Please note that your current contact point is a co-author on this manuscript. According to our Data Policy, the contact point must not be an author on the manuscript and must be an institutional contact, ideally not an individual. Please revise your data statement to a non-author institutional point of contact, such as a data access or ethics committee, and send this to us via return email. Please also include contact information for the third party organization, and please include the full citation of where the data can be found.

Response: We appreciate the editor’s comment and reviewed the data access statement. Now data and supplementary materials are publicly available and included in the submission.

Response: We appreciate the editor’s suggestion. We have now included a full ethics statement in the Methods section, specifying the name of the IRB and institution that approved the study, consent procedures, and data confidentiality. Please see page 8, lines 156-162 of the revised manuscript

“Ethical clearance was obtained from the Institutional Review Board of Texas Tech University (Approval no. IRB2021-972). The study protocol and survey instrument were reviewed and approved prior to data collection. Participants provided informed consent electronically before beginning the survey, and they were informed that participation was voluntary, responses would remain anonymous, and they could withdraw at any time. Data were collected from January 24, 2022, until June 17, 2022, and analyzed shortly after data collection was completed. All identifiers were removed before the formal analysis.”

Response: We thank the editor for the guidance on supporting information. We have included a Supporting Information section at the end of the manuscript, identifying and labeling the supplementary tables according to PLOS ONE formatting guidelines. Please see page 32, lines 639-645 of the revised manuscript. Specifically, we added “S1 Table. Mixed logit model estimation results” and “S2 Table. Random-effects regression analysis of sociodemographic characteristics.”

Response: We appreciate the editor’s clarification on this topic. The reviewers did not request specific previously published works. However, we reviewed the relevant literature regarding the reviewers’ comments and added references to strengthen the revised manuscript to address the comments.

Response: We address the editor’s comment for the reference list. We made sure that the references follow the journal guidelines.

Reviewer #1

The authors have addressed a very timely topic in this manuscript, at a time when nutritional challenges such as high fructose corn syrup (HFCS) are increasingly recognized as detrimental for health. They have supported the consumer perception, consumption pattern, and valuations with crisp data.

Response: We would like to thank the reviewer for taking the time to review the manuscript and the positive comments. We are pleased that the reviewer found our research well supported.

The following comment may be incorporated to improve the manuscript further:

When discussing about concerns due to consumption of HFCS, the authors may include a statement on HFCS in relation to FODMAPs and their impact on GI motility disorders such as IBS, post prandial distress, as well as inflammatory conditions such as IBD. These conditions afflict a lot of people around the world.

Response: We thank the reviewer for this valuable suggestion, which helped strengthen the justification for studying HFCS and provides a clearer context of the health implications of HFCS consumption. We included the statement in the introduction section describing the relationship. Please see page 4, lines 67-74 of the revised manuscript

“While all sweeteners have potential health implications, the primary difference between HFCS and sugar is their fructose content. HFCS is often considered a high-FODMAP sweetener because it contains a higher proportion of fructose relative to glucose. Excess fructose can be rapidly absorbed by the body and has been associated with inflammation, insulin resistance, increased fat production, and non-alcoholic fatty liver disease [20]. In addition, many individuals may have difficulty absorbing excess fructose, which can worsen symptoms such as bloating, abdominal pain, or post-meal discomfort, and may contribute to inflammatory conditions, including inflammatory bowel disease [21-23].”

Reviewer #2

This study explores the links between high-fructose corn syrup (HFCS) consumption, public perception, and labeling preferences for popular snacks. The findings offer valuable insights for food manufacturers, policymakers, and health educators working to address the divide between the science of HFCS and consumer views. The methods are well-designed, and the results are clearly presented, contributing meaningful understanding to this often-debated ingredient.

Response: We would like to thank the reviewer for taking the time to review the manuscript and the positive comments. We appreciate the acknowledgment of the relevance of this topic and the contribution of our findings to understanding consumer perceptions and labeling preferences regarding HFCS.

Finally, we thank the editor and reviewers for their time and thoughtful feedback. We hope that our revised manuscript satisfactorily addresses their comments and meets the publication criteria of PLOS ONE.

---

## [Editor Report · Decision Letter 1]

11 Jan 2026

Understanding High Fructose Corn Syrup in Popular Snacks: Consumption, Perceptions and Labeling Preferences

PONE-D-25-45729R1

Dear Dr. Sarasty,

We’re pleased to inform you that your manuscript has been judged scientifically suitable for publication and will be formally accepted for publication once it meets all outstanding technical requirements.

Kind regards,

Edwin Hlangwani, PhD

Academic Editor

PLOS One
---

## [Editor Report · Acceptance letter]

PONE-D-25-45729R1

PLOS One

Dear Dr. Sarasty,

I'm pleased to inform you that your manuscript has been deemed suitable for publication in PLOS One. Congratulations! Your manuscript is now being handed over to our production team.

Kind regards,

on behalf of

Dr. Edwin Hlangwani

Academic Editor

PLOS One